# Predicting hygroscopic growth of organosulfur aerosol particles using COSMO*therm*

Zijun Li[1], Angela Buchholz[2], and Noora Hyttinen[3,a]

[1]International Laboratory for Air Quality and Health, School of Earth and Atmospheric Sciences, Queensland University of Technology, Brisbane QLD 4001, Australia
[2]Department of Technical Physics, University of Eastern Finland, Kuopio FI-70210, Finland
[3]Department of Chemistry, Nanoscience Center, University of Jyväskylä, Jyväskylä FI-40014, Finland
[a]now at: Atmospheric Research Centre of Eastern Finland, Finnish Meteorological Institute, FI-70211, Kuopio, Finland

**Correspondence:** Zijun Li (zijun.li@qut.edu.au) and Noora Hyttinen (noora.hyttinen@fmi.fi)

**Abstract.** Organosulfur (OS) compounds are important sulfur species in atmospheric aerosol particles, due to the reduction of global inorganic sulfur emissions. Understanding the physicochemical properties, such as hygroscopicity, of OS compounds is important for predicting future aerosol-cloud-climate interactions. However, their hygroscopicity is not yet well understood due to the scarcity of authentic standards. In this work, we investigated a group of OS compounds with short carbon chains ($C_1$-$C_5$) and oxygen-containing functional groups in the form of sodium, potassium, or ammonium salts, and their mixtures with ammonium sulfate. The hygroscopic growth factors (HGF) of these OS compounds have been experimentally studied. Here, the HGFs were calculated from water activities computed using the conductor-like screening model for real solvents (COSMO-RS). A good agreement was found between the model-estimated and experimental HGFs for the studied OS compounds. This quantum-chemistry-based approach for HGF estimation will open up the possibility of investigating the hygroscopicity of other OS compounds present in the atmosphere.

## 1 Introduction

Atmospheric aerosol particles affect the global climate directly by scattering solar radiation or indirectly by seeding clouds. As important aerosol constituents, organosulfur (OS) compounds contribute up to 30% of total organic aerosol particles in mass (Surratt et al., 2008; Tolocka and Turpin, 2012). Typically, these OS compounds contain a sulfonate or sulfate ester group. Methanesulfonic acid, the most abundant OS compound in marine environments, can be produced via both gas and aqueous-phase oxidation of reduced-sulfur compounds (Barnes et al., 2006; Hoffmann et al., 2016; Berndt et al., 2023). Another OS compound of high abundance (Moch et al., 2020), namely hydroxymethanesulfonate, is formed from sulfite and formaldehyde in the aqueous phase (Boyce and Hoffmann, 1984; Song et al., 2019). Moreover, other OS compounds can be formed via multi-phase oxidation of biogenic as well as anthropogenic organic compounds in the presence of sulfuric acid. To date, OS compounds have been identified in various environments across forest (Iinuma et al., 2007; Kristensen and Glasius, 2011), coastal (Huang et al., 2015; Zhou et al., 2023), polar (Hansen et al., 2014; Ye et al., 2019; Campbell et al., 2022), and urban

sites (Jiang et al., 2022; Glasius et al., 2022). Therefore, the abundance and ubiquity of OS compounds in the atmosphere highlight their importance for global climate.

Understanding the interactions of OS compounds with water vapor as a function of relative humidity (RH) is important in assessing their climate effect. When in equilibrium with surrounding RH, the degree of water uptake (i.e., hygroscopicity) largely affects aerosol phase states, optical properties, chemical reactivity, and cloud formation potential. A range of laboratory studies have investigated the hygroscopicity for both commercially available and synthesized OS compounds under sub-saturated (RH<100%) conditions (Hansen et al., 2015; Estillore et al., 2016; Peng et al., 2021, 2022; Ohno et al., 2022; Bain et al., 2023). Inorganic salts typically show distinct deliquescence and efflorescence RH (DRH and ERH, respectively) points, while the studied OS compounds consistently show continuous growth with increasing RH. Furthermore, the presence of OS compounds can lower the DRH and ERH of inorganic salt particles (Estillore et al., 2016; Peng et al., 2021, 2022), and potentially extends the RH range where particulate water is present and thus influences the physicochemical properties of atmospheric aerosol particles.

Both Aerosol Inorganic-Organic Mixtures Functional Groups Activity Coefficients (AIOMFAC) (Zuend et al., 2008, 2011; Zuend and Seinfeld, 2012) and conductor-like screening model for real solvents (COSMO-RS) (Klamt, 1995; Klamt et al., 1998; Eckert and Klamt, 2002) can estimate the activity coefficients of compounds present in atmospheric aerosol particles as a function of RH. Previous thermodynamic studies have shown that both AIOMFAC and COSMO-RS can provide similar predicted water activities ($\alpha_w$) in aqueous $(NH_4)_2SO_4$, $NH_4HSO_4$, $NH_4NO_3$, and $NH_4IO_3$ (Hyttinen, 2023a) as well as in carboxylic acids (Hyttinen et al., 2020b; Hyttinen and Prisle, 2020). The group contribution calculations in AIOMFAC can be finished in seconds, while the quantum chemistry calculations in COSMO-RS require hours to be completed. However, COSMO-RS is the only existing method able to estimate $\alpha_w$ in OS solutions. Recently, the solubilities and activities of isoprene- and monoterpene-derived organosulfates were computed using COSMO-RS (Hyttinen et al., 2020a). To our knowledge, there is, however, no comparison between the experimental and thermodynamic model-estimated hygroscopicity of OS compounds, which become increasingly important due to declining $SO_2$ emissions (Riva et al., 2019). This hinders the understanding of their atmospheric impacts and fates.

Here, we perform quantum chemistry calculations with COSMO-RS to explore a set of atmospherically relevant OS compounds (Table S1 in the Supplement), including

– Sodium OS: methyl sulfate (NaMS), hydroxymethanesulfonate (NaHMS), ethyl sulfate (NaES), and 2-hydroxyethylsulfonate (NaHES)

– Potassium OS: glycolic acid sulfate (KGAS), hydroxyacetone sulfate(KHAS), 2-butenediol sulfate (KBS), and 4-hydroxy-2,3-epoxybutane sulfate (KHEBS)

– Ammonium OS: 2-hydroxyethyl-sulfonate ($NH_4HES$), (2R,3S)-1,3,4-trihydroxy-2-methylbutan-2-yl sulfate ($NH_4TMS$ (a)), and (2R,3R)-2,3,4-trihydroxy-2-methylbutan-2-yl sulfate ($NH_4TMS$ (b)).

The experimental hygroscopicities of these OS are reported in the literature (Estillore et al., 2016; Peng et al., 2021, 2022; Ohno et al., 2022). For each studied OS, we estimate the corresponding $\alpha_w$ at a range of solute concentrations to predict the

hygroscopic growth curves. Additionally, we predict particle hygroscopicity for the OS mixed with ammonium sulfate (AS; $(NH_4)_2SO_4$), which is the most abundant inorganic salt in atmospheric aerosol particles.

## 2 Computational methods

### 2.1 Activity coefficients

The COSMO-RS model (Klamt, 1995; Klamt et al., 1998; Eckert and Klamt, 2002), implemented in the BIOVIA COSMO*therm* program (BIOVIA COSMO*therm*, 2021) (abbreviated COSMO*therm*), was used to calculate activity coefficients of water under vapor-liquid equilibrium. All activity coefficients were calculated using the most recent BP_TZVPD_FINE_21 parametrization (abbreviated FINE), with the exception of solutions containing AS, which were calculated using the newly developed electrolyte parametrization BP_TZVP_ELYTE_21 (abbreviated ELYTE; see Section S1 in the Supplement for more details). The activity coefficient $\gamma_j$ of a compound $j$ is computed using the pseudo-chemical potential $\mu^*$ (Ben-Naim, 1987) at composition $\mathbf{x}$ and at the reference state $\mathbf{x}^\circ$:

$$\ln\gamma_j(\mathbf{x}) = \frac{\mu_j^*(\mathbf{x}) - \mu_j^{*,\circ}(\mathbf{x}^\circ, T, P)}{RT} \tag{1}$$

where $T$ is the temperature (295 K), $R$ is the gas constant (in kJ $K^{-1}mol^{-1}$) and $P = 10^5$ Pa is the reference pressure. The pseudo-chemical potential $\mu_j^*$ is an auxiliary quantity defined using the chemical potential at the reference state $\mu^\circ$:

$$\mu_j^*(\mathbf{x}) = \mu_j{}^\circ(\mathbf{x}^\circ, T, P) + RT\ln\gamma_j(\mathbf{x}) \tag{2}$$

Pure water with mole fraction ($x_w$) of 1 is used as the reference state composition $\mathbf{x}^\circ$. With the calculated $\gamma_j$, each aqueous-phase composition is paired with a RH assuming that at equilibrium, $\alpha_w$ = RH/100%, where $\alpha_w = x_w\gamma_w$.

### 2.2 Input files for COSMO*therm* calculations

Input files for COSMO*therm* calculations (cosmo-files) were obtained through a series of density functional theory calculations with increasing levels of theory. The process has been discussed in more detail in a previous publication (Hyttinen et al., 2020a). In short, all conformers were found using the systematic conformer search algorithm in the Spartan20 program (Wavefunction Inc., 2020). The geometries of all conformers were optimized and duplicate conformers were removed using the BIOVIA COSMO*conf* program (BIOVIA COSMO*conf*, 2021; TURBOMOLE, 2020). The final cosmo-files were computed at the BP/def2-TZVPD-FINE//BP/def-TZVP level of theory (BP/def-TZVP for BP_TZVP_ELYTE_21 calculations).

Many of the studied cations have multiple conformers. At most 10 lowest chemical potential conformers were selected as inputs for the COSMO*therm* calculations. However, only conformers with chemical potentials within 8 kJ mol$^{-1}$ of the lowest chemical potential were used, in order to avoid including conformers with low COSMO energies but high chemical potentials (Hyttinen, 2023b). More specifically, COSMO*therm* gives high weights to conformers containing intramolecular H-bonds (Hyttinen and Prisle, 2020), because intramolecular H-bonds are favored in the COSMO energies (Kurtén et al., 2018).

## 2.3 Predicting HGF of OS particles

The hygroscopicity of an organic compound is typically quantified using the hygroscopic growth factor (HGF), which is the ratio of the diameter at a RH condition $i$ ($D_{p,i}$) to the diameter at dry conditions (RH$\leq$10 %; $D_{p,0}$):

$$\text{HGF} = \frac{D_{p,i}}{D_{p,0}} \tag{3}$$

Assuming particle sphericity, HGF can be further expressed with:

$$\text{HGF} = \left(\frac{V_{p,i}}{V_{p,0}}\right)^{\frac{1}{3}} \tag{4}$$

where $V_{p,i}$ and $V_{p,0}$ are the particle volumes at RH condition $i$ and dry conditions, respectively. Assuming volume additivity, HGF can be represented using the volume of water and the solute ($V_{OS}$ and $V_{H2O}$).

$$\text{HGF} = \left(\frac{V_{\text{OS}} + V_{\text{H}_2\text{O},i}}{V_{\text{OS}} + V_{\text{H}_2\text{O},0}}\right)^{\frac{1}{3}} \tag{5}$$

The volume of each component $j$ ($V_j$) can be written using the mass at RH = $i$ ($m_{j,i}$) and density $\rho_j$:

$$V_j = \frac{m_{j,i}}{\rho_j} \tag{6}$$

In COSMO*therm*, $\rho_j$ is estimated from the molecular volumes of individual species present in the mixture, assuming close packing. Therefore, the formation of intermolecular H-bonds within the mixture is not considered in the density calculation.

Combing Eqs. 5 and 6 leads to:

$$\text{HGF} = \left(\frac{\frac{m_{\text{OS}}}{\rho_{\text{OS}}} + \frac{m_{\text{H}_2\text{O},i}}{\rho_{\text{H}_2\text{O}}}}{\frac{m_{\text{OS}}}{\rho_{\text{OS}}} + \frac{m_{\text{H}_2\text{O},0}}{\rho_{\text{H}_2\text{O}}}}\right)^{\frac{1}{3}} = \left(\frac{1 + \frac{m_{\text{H}_2\text{O},i} \cdot \rho_{\text{OS}}}{m_{\text{OS}} \cdot \rho_{\text{H}_2\text{O}}}}{1 + \frac{m_{\text{H}_2\text{O},0} \cdot \rho_{\text{OS}}}{m_{\text{OS}} \cdot \rho_{\text{H}_2\text{O}}}}\right)^{\frac{1}{3}} \tag{7}$$

where the mass ratio between $m_{\text{H}_2\text{O}}$ and $m_{\text{OS}}$ can be predicted using the COSMO*therm*-estimated mass fraction of water at equilibrium. This approach treats the OS particles as a bulk phase without considering particle size.

## 2.4 Predicting HGF of OS-AS mixture particles

COSMO*therm* calculations assume that all salts are dissolved in water regardless of the water content. However, AS may not fully dissolve in aqueous mixtures under low RH conditions. Previous experimental studies show that when AS is mixed with certain carboxylic acids (Choi and Chan, 2002; Chan et al., 2006) and OS (Peng et al., 2021, 2022), a step change in HGF was typically observed at or below the DRH of AS, which is attributed to the full dissolution of AS. In other words, at RH below the observed step change in HGF, AS may exist in a crystalline solid state or partially dissolve in the liquid phase. For all the studied OS-AS mixture particles, step-wise changes in HGFs were observed in the measurements conducted by Peng et al. (2021, 2022). Similar to Hodas et al. (2016), which studied the HGF of polyethylene glycol oligomers mixed with AS, we assume that AS exists only in the solid state before reaching full deliquescence in the calculations. When the RH increases above the DRH, AS reaches its solubility limit and undergoes a solid-to-liquid phase transition. At the DRH, the molar ion

activity product (IAP) of the inorganic salt (e.g., AS) will be the same, regardless of other components in the mixture. It is therefore possible to determine the DRH of different salt mixtures, if the IAP of the inorganic salt for one saturated solution (IAP$_{sat}$) can be calculated.

For a specific ion, the molal activity coefficient $\gamma_{j,b}$ can be computed using the mole fraction-based activity coefficient $\gamma_{j,x}$ estimated by COSMO*therm* (Robinson and Stokes, 2002):

$$\gamma_{j,b} = \frac{\gamma_{j,x}}{1 + 0.001 \cdot MW \cdot \Sigma_j \nu_j b_j} \tag{8}$$

where $MW$ is the molar mass of the solvent water, $\nu_j$ is the number of moles of ions formed by the ionization of one mole of salt $j$ (3 for AS and 2 for OS) and $b_j$ is the molality (i.e., moles of solute per kg of water) of salt $j$. The IAP of AS in a solution can be calculated using the molal ionic activity coefficients and molalities of $NH_4^+$ and $SO_4^{2-}$ (Robinson and Stokes, 2002):

$$IAP = [\gamma_{NH_4^+,b} \cdot b_{NH_4^+}]^2 [\gamma_{SO_4^{2-},b} \cdot b_{SO_4^{2-}}]^1 \tag{9}$$

The IAP$_{sat}$ is calculated using the aqueous solubility limit of AS (i.e., 5.790 mol AS per kg of water at 295 K). When the estimated IAP>IAP$_{sat}$, AS is assumed to exist only in its solid form. In this case, the OS is the sole component contributing to the particle water uptake. Under such a condition, we apply the Zdanovskii–Stokes–Robinson (ZSR) approach Stokes and Robinson (1966) to estimate the particle water uptake, assuming no interactions between AS and the liquid phase. The corresponding HGF of the OS-AS mixture is derived by multiplying the COSMO*therm*-estimated HGF for the pure OS with the OS volume fraction in the OS-AS mixture. When the estimated IAP $\leq$ IAP$_{sat}$, AS is assumed to be fully dissolved in the liquid phase and well mixed with OS. Similar to Eq. 7 which treats the particles as a bulk phase, the HGF in this case can be derived based on the COSMO*therm*-estimated $\alpha_w$ of the OS-AS mixture, with the aid of Eq. 10 expressed as below:

$$HGF = \left( \frac{\frac{m_{OS}}{\rho_{OS}} + \frac{m_{AS}}{\rho_{AS}} + \frac{m_{H_2O,i}}{\rho_{H_2O}}}{\frac{m_{OS}}{\rho_{OS}} + \frac{m_{AS}}{\rho_{AS}} + \frac{m_{H_2O,0}}{\rho_{H_2O}}} \right)^{\frac{1}{3}} = \left( \frac{1 + \frac{m_{AS} \cdot \rho_{OS}}{m_{OS} \cdot \rho_{AS}} + \frac{m_{H_2O,i} \cdot \rho_{OS}}{m_{OS} \cdot \rho_{H_2O}}}{1 + \frac{m_{AS} \cdot \rho_{OS}}{m_{OS} \cdot \rho_{AS}} + \frac{m_{H_2O,0} \cdot \rho_{OS}}{m_{OS} \cdot \rho_{H_2O}}} \right)^{\frac{1}{3}} \tag{10}$$

where $m_{AS}$ and $\rho_{AS}$ are the mass and density of AS.

## 3   Results and discussion

### 3.1   Water activity of OS

$\alpha_w$ of the studied OS were estimated in solutions with water mass fractions ($m_w$) ranging from 0 to 0.96 (Fig.1). Since different OS have different molecular weights, using $m_w$ enables a direct comparison of the water uptake for the studied salts. All OS are assumed to be fully dissolved in the water as ions. Due to the scarcity of experimental data on $\alpha_w$ for the studied OS, it is not possible to determine the relative errors of the model estimates for all studied OS. Compared with the only existing measured $\alpha_w$ data from Bain et al. (2023), COSMO*therm* calculations provided similar values of $\alpha_w$ for NaMS and NaES (Fig. S2 in the Supplement).

In the sodium OS group, the studied OS with the same carbon number exhibit higher $m_w$ with more oxygenated functional groups at a fixed $\alpha_w$ below 0.7 (i.e., RH below 70%). Here, COSMO*therm* predicts higher $\alpha_w$ in more concentrated solutions and lower $\alpha_w$ in more diluted solutions for NaMS and NaES, compared with those containing the isomeric hydroxy sulfonate salts (i.e., NaHMS and NaHES). This is likely caused by the relatively more hydrophobic nature of the methyl and ethyl groups compared to anions that contain hydrogen bond donating functional groups (e.g., -OH groups). Similar but stronger patterns in $\alpha_w$ are seen in the unstable regions of phase-separating mixtures (Hyttinen, 2023b). In the potassium OS group, the selected OS display higher $m_w$ with increasing molecular weights at a fixed $\alpha_w$. The three chosen ammonium OS show similar degrees of water uptake, independent of the anion. When comparing the three cation groups (i.e., Na$^+$, K$^+$, and NH$_4$$^+$), we observed that at a fixed $\alpha_w$, the group of potassium OS (Fig.1b) shows the lowest degree of water uptake as indicated by the lowest $m_w$. Note that most of the anions are different between the three cation groups. The difference in $\alpha_w$ between the three cation groups can arise from the differences in cations and/or anions.

To rule out the effect of cations, we additionally computed the $\alpha_w$ for all cation-anion pairs (Fig. S3 in the Supplement). Regardless of the anion, at any $\alpha_w$, potassium OS show the lowest $m_w$, as compared to the corresponding sodium and ammonium OS. Note that at $\alpha_w \leq 0.7$, most sodium OS have lower equilibrium water content compared to the corresponding ammonium OS. However, at $\alpha_w > 0.7$, both sodium and ammonium OS show similar equilibrium water content for each of the studied anions. Moreover, for each studied OS, the order of cations is more or less the same, suggesting the presence of the Hofmeister effect. Sodium is usually enriched in sea spray particles (Salter et al., 2016), while potassium and ammonium are commonly found in continental aerosol particles influenced by biomass burning (Vasilakopoulou et al., 2023) and anthropogenic ammonia sources (Pai et al., 2021). Therefore, the $\alpha_w$ of OS in aerosol particles may vary depending on the influence of marine and continental air masses.

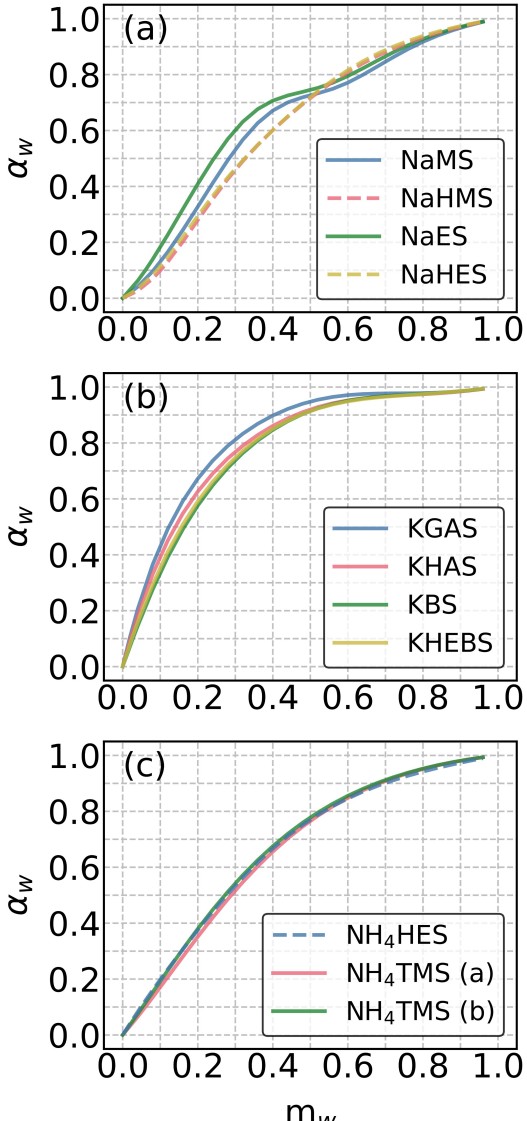

**Figure 1.** COSMO*therm*-derived water activities ($\alpha_w$) in aqueous solutions of the studied (a) sodium, (b) potassium, (c) and ammonium OS as a function of water mass fraction ($m_w$) at 295 K. The solid and dashed lines represent salts of organosulfates and organosulfonates, respectively.

## 3.2 Hygroscopicity of OS particles

We calculated the HGF for each studied OS, on the basis of the COSMO*therm*-estimated $m_w$ and density ($\rho$, Table S1 in the Supplement). The corresponding COSMO*therm*-estimated HGFs with the experimental data are shown for three selected OS (i.e., NaMS, KGAS, and NH$_4$HES) particles in Fig. 2. Both the measured and computed HGFs are very similar to each other.

Fig. S4 in the Supplement summarizes the experimental and computed HGF data of all the studied OS particles as a function of RH. To evaluate the model performance for the studied OS particles, we compared the model simulations against the measurement data as indicated by the relative differences shown in Fig. S5 in the Supplement. There are different possible sources of uncertainties that may explain the discrepancy between experimental data and COSMO*therm* estimates. The uncertainty can originate from the errors in the experiments, the COSMO*therm*-estimated $\alpha_w$, or the assumptions made for calculating the HGF from $\alpha_w$. In this study, most of the discrepancies likely originate from the COSMO*therm*-estimated $\alpha_w$ of some of the ions. The original COSMO*therm* parametrization (Klamt et al., 1998) provided only poor estimates for water. The most recent COSMO*therm* parametrization performs much better and we found relatively good agreement between experimental data and our HGF calculations. Note that the COSMO*therm*-predicted HGFs overall agree very well with the measurement data, mostly showing relative differences of $\pm 5\%$ or less against the measured HGFs (Fig. S5). This agreement highlights the validation of using $m_w$ and $\rho$ from COSMO*therm* to predict the hygroscopic growth of OS particles.

In addition, we compared the HGF data using all three available parametrizations of the COSMO-*therm* program (FINE, TZVP, and ELYTE), as shown in Fig. S6 in the Supplement. For the four studied sodium OS particles, ELYTE gave similarly good or even better HGF estimates, compared to FINE. For the potassium OS particles, the FINE HGF estimates show the best agreement with the measurement data, compared to the TZVP and ELYTE estimates. When considering the three studied ammonium OS particles, the three parametrizations provided similar RH-dependency of HGFs but gave only reasonably good HGF estimates for NH$_4$HES particles. Among the three COSMO*therm* parametrizations, FINE overall provided the best agreement across all the studied OS particles and showed the smallest relative differences against the measurement data (Fig. S5). Therefore, FINE was chosen for the detailed analysis here.

We acknowledge that the effect of surface tension was not taken into account in the HGF calculation for each studied OS. However, considering the dry OS particle size of 100 nm or larger from the experimental studies, the surface tension effect can be assumed to be negligible in the sub-saturated regime (Bezantakos et al., 2016). For smaller particles and surface-active compounds, the surface tension may significantly affect the equilibrium water content.

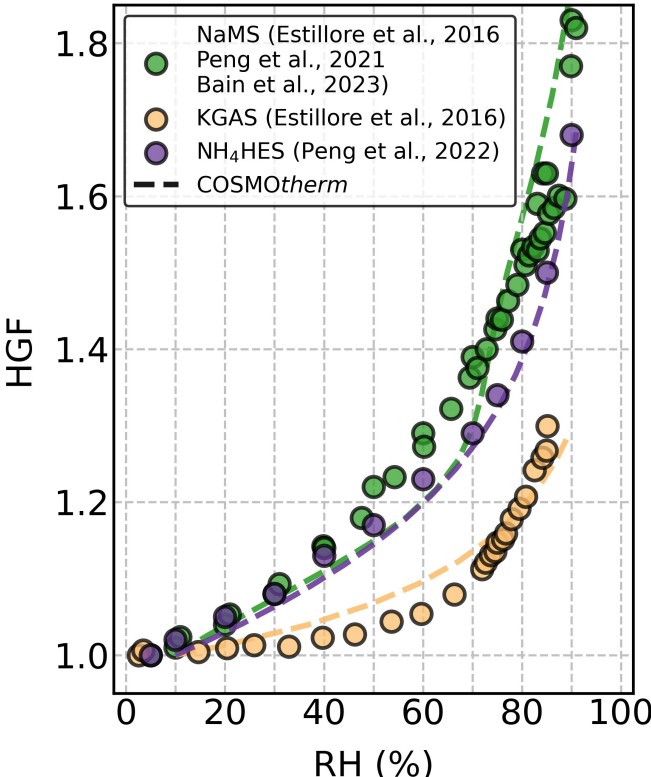

**Figure 2.** Hygroscopic growth factors (HGFs) for sodium methyl sulfate (NaMS; green), potassium glycolic acid sulfate (KGAS; yellow), and ammonium 2-hydroxyethyl-sulfonate (NH$_4$HES; purple) as a function of RH at 295 K. The HGF data from the literature (Estillore et al., 2016; Peng et al., 2021, 2022; Bain et al., 2023) and the COSMO*therm*-derived calculations are shown in solid circles and dashed lines, respectively.

### 3.3 Hygroscopicity of OS-AS mixture particles

We also examined the RH dependency of HGF for the studied OS-AS mixture particles at mass ratios (OS:AS) of 1:1, 1:3, and 1:5. Fig. 3 shows the ZSR-predicted HGF as a function of RH for the three methyl sulfate (MS) salts mixed with AS. Whenever available, the corresponding experimental HGF data are presented as well. For other OS-AS mixture particles, the HGF data are presented in Fig. S7 in the Supplement.

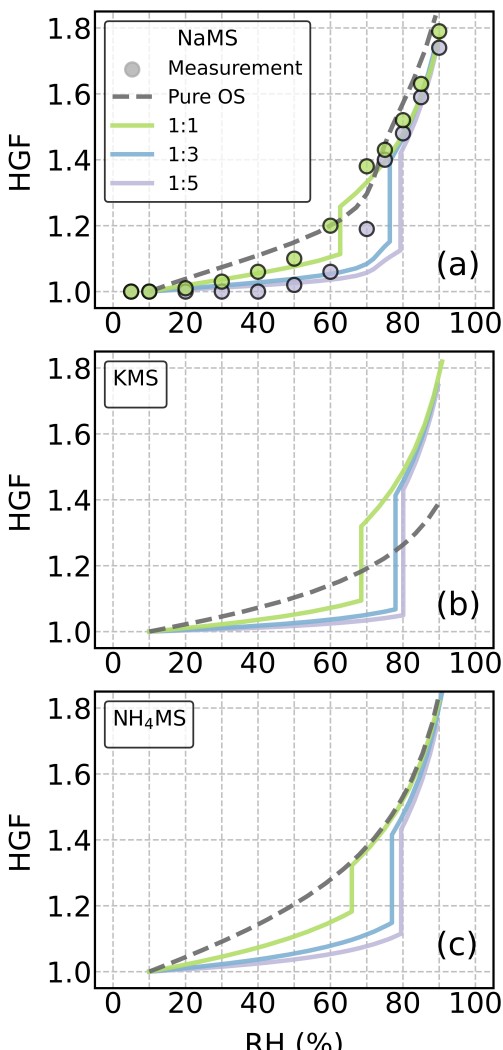

**Figure 3.** Hygroscopic growth factors (HGFs) of the mixture particles of ammonium sulfate (AS) and (a) sodium, (b) potassium and (c) ammonium methyl sulfate as a function of RH at 295 K. The filled circles indicate measurement data from the literature (Peng et al., 2021). The HGF data derived from the COSMO*therm*-based calculations are in dashed lines for pure organosulfur (OS) compounds and in solid lines for the OS-AS mixed particles. Colors indicate different OS:AS mass ratios.

The estimated HGF show that unlike the pure MS particles (grey dashed lines in Fig. 3), almost all the MS-AS mixture particles (solid lines) with the three chosen mass ratios exhibited gradual water uptake behaviours and then sharp deliquescence transitions at 60 % RH or above. With an increasing mass fraction of AS in the solute, the DRH of the MS-AS mixture particles shifts to higher RH but is still lower than the DRH of the pure AS (i.e., 80% RH). The presence of AS noticeably lowered water uptake compared to the pure OS cases when RH was below the DRH. Similar results were observed in other OS-AS mixture

particles (Fig. S7). However, when RH was above the DRH, the addition of AS only increases water uptake of KMS and other potassium OS particles.

Our HGF estimates are able to reproduce the measured HGF curves of the NaMS-AS mixture particles with 1:1 mass ratio (Fig. 3a). For the NaMS-AS mixture particles with 1:5 mass ratio, the HGF estimate well predicts the measurement data at RH < 60% or RH > 80% but there is disagreement around the DRH. Similar HGF underestimation at 60%–80% RH was also observed in NaHMS, NaHES, NaES and $NH_4HES$ mixture particles with AS with 1:3 and 1:5 mass ratios. For these 1:3 and 1:5 OS-AS mixture particles, the discrepancy between the measured and estimated HGFs is likely due to the underestimated interaction between OS and AS near the deliquescence phase transition and the resultant overestimated IAP of AS. It is also possible that solid AS partially dissolves into the liquid phase before the onset of deliquescence, thus leading to higher HGF at 60%–80% RH than expected. Such partial dissolution behavior of AS is not accounted for in our approach when the IAP of AS is below $IAP_{sat}$.

## 4    Conclusions

This novel approach does not require any reference OS data for optimization prior to HGF estimations. Instead, it is solely based on the quantum chemistry calculations and existing parametrizations of the commercially available COSMO*therm* program. In this study, the COSMO*therm* calculations show how the presence of sodium, potassium, and ammonium affects the $\alpha_w$ of OS. Given that these three cations originate from different emission sources in continental and marine environments, the $\alpha_w$ of OS can vary from one place to the next. Previous thermodynamic studies using COSMO*therm* have provided $\alpha_w$ estimates consistent with those measured in bulk and particle phases for multifunctional atmospheric organics (Hyttinen et al., 2020b; Hyttinen and Prisle, 2020). As the first study modeling HGFs based on the COSMO*therm*-estimated $\alpha_w$, we show good agreement between COSMO*therm*-derived and experimental HGFs in most cases of the studied OS. This highlights the potential applicability of COSMO*therm* for estimating HGFs for other atmospheric compounds in future works.

Globally, $SO_2$ emissions are projected to decrease due to the phasing out of fossil fuels. This is expected to decrease the total and inorganic sulfur, thereby increasing the contribution of OS to total sulfur in atmospheric aerosol particles (Riva et al., 2019; Brüggemann et al., 2020). With progress in analytical instruments and methods, hundreds of OS compounds have been recently identified from field measurements (Hettiyadura et al., 2017, 2019; Wang et al., 2021; Huang et al., 2023; Wang et al., 2023; Yang et al., 2023). However, investigation of the physicochemical properties of OS is still lacking due to the scarcity of authentic standards. Our COSMO*therm*-based approach allows characterizing the hygroscopicity of OS particles even when authentic standards are unavailable. This will help us understand the aerosol-cloud-climate interactions in the post-fossil-fuel future where OS compounds are highly important.

*Data availability.* The data set is available upon request from the corresponding author.

*Author contributions.* ZL and NH conceived the study. ZL performed the data collection and hygroscopicity calculation. NH performed the COSMO*therm* calculations. ZL, AB, and NH analyzed and interpreted data. ZL wrote the paper with contributions from all coauthors.

*Competing interests.* The author has declared that there are no competing interests.

230 *Acknowledgements.* ZL thanks the QUT Early Career Research Scheme for funding support. NH thanks the Research Council of Finland, Grant No. 338171, for the financial contribution and CSC–IT Center for Science, Finland, for the computational resources. We thank Dr Henry Oswin for the helpful discussion.

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
