# Peer review of "Predicting hygroscopic growth of organosulfur aerosol particles using COSMO*therm"

_EGUsphere, 2024_

## Author Comment (AC1)

We thank all reviewers for their insightful comments. Below, we provide point-by-point responses to each comment. In the following text, the **reviewers' comments and suggestions** are in black, **authors' responses** are in red, and **changes to the manuscripts and supplement information** are in blue. Additionally, we corrected typos and grammatical errors in the manuscript and supplement that previously were unnoticed.

**R1    Response to Reviewer 1**

**R1.1    General Comments**

This manuscript titled "Predicting Hygroscopic Growth of Organosulfur Aerosol Particles Using COSMOtherm" by Zijun Li et al. estimated the HGF of OS compounds and their mixtures with ammonium sulfate using the conductor-like screening model (COSMOtherm) and compared the results with those from existing experimental studies. It claimed that the model-estimated and experimental HGFs for the studied OS compounds agreed well, based on which it proposed that the quantum-chemistry-based approach for HGF estimation will open up the possibility of investigating the hygroscopicity of other OS compounds present in the atmosphere. The addressed scientific question is well within the scope of ACP and the manuscript is well structured and well-written.

However, more explanation of the COMSMOtherm model and the appropriateness of using it in the study should be included. E.g, how this model has successfully estimated the water activity of other organic compounds other than OS? What are the merits of this model compared with other models in estimating the water activity of OS and mixtures of OS and AS? In addition, more discussions in Sect. 3 are necessary to help readers better understand the results obtained. Please refer to the specific comments for details.

Response: Thank you for your insightful comments. We addressed each specific comments as follows.

**R1.2    Specific comments**

1. Line 5: "a group of compounds", more specific explanations on the OS compounds could benefit the readers.

   Response: Thanks for your suggestion. We revised the sentence by providing more specific explanations for the chosen OS.

   Change: Abstract
   [...] In this work, we investigated a group of OS compounds with short carbon chains ($C_1$-$C_5$) and oxygen-containing functional groups in the form of sodium, potassium, or ammonium salts, and their mixtures with ammonium sulfate. We selected OS compounds for which the hygroscopic growth factors (HGF) have been experimentally studied. [...]

2. Lines 33-36: what are the respective merits and demerits of AIOMFAC and COSMO-RS? Why did you choose to use COSMO-RS? Is there any existing study using AIOMFAC?

   Response: Both AIOMFAC and COSMO-RS can provide highly precise estimations of water activities ($\alpha_w$) for ranges of atmospherically relevant species. Previous thermodynamic studies have shown that these two models can predict similar $\alpha_w$ in aqueous solutions containing $(NH_4)_2SO_4$, $NH_4HSO_4$, $NH_4NO_3$, or $NH_4IO_3$ (Hyttinen, 2023a) and in those containing mono- and dicarboxylic acids (Hyttinen et al., 2020).

   AIOMFAC is a contribution-based model that offers a wide variety of functional groups (Zuend et al., 2008, 2011; Zuend and Seinfeld, 2012). Typically, AIOMFAC calculations can be completed within a timescale of seconds. Unfortunately, organosulfur (OS) compounds have not yet been parameterized into AIOMFAC due to the lack of experimental $\alpha_w$ data of aqueous OS solutions. Recently, Bain et al. (2023) found a good agreement between the measured $\alpha_w$ for sub-saturated sodium methyl and ethyl sulfate and the AIOMFAC prediction for sodium sulfate and bisulfate. However,

it remains uncertain if the sulfonate or sulfate ester groups in other OS compounds can be treated as sulfate or bisulfate groups in AIOMFAC.

Different from AIOMFAC, COSMO-RS is a quantum chemistry model. It uses quantum chemistry and statistical thermodynamics to estimate $\alpha_w$ in aqueous solutions containing inorganic or organic compounds. To our knowledge, COSMO-RS is the only method capable of estimating $\alpha_w$ in aqueous OS solutions. Compared with AIOMFAC, the COSMO-RS calculations are much slower (e.g., several hours), primarily due to the quantum chemistry input required for the COSMO-RS calculations. Compared with the only existing measured $\alpha_w$ data from Bain et al. (2023), COSMO*therm* provided similar $\alpha_w$ estimates for NaMS and NaES (Fig. R1).

To provide more information about AIOMFAC and COSMO-RS and about the performance of COSMO-RS, now we include a short description of both models in the Introduction part and Section 3.1, and also update Fig. 1 in the main text and Figs. S3 and S4 in the Supplement by including the HGF data from Bain et al. (2023).

[Figure]

. Figure R1. Comparisons between the COSMO*therm*-predicted water activities ($\alpha_w$) and literature data (Bain et al., 2023) for NaMS and NaES.

Change:
Introduction
[...] Previous thermodynamic studies have shown that both AIOMFAC and COSMO-RS can provide similar predicted water activities ($\alpha_w$) in aqueous $(NH_4)_2SO_4$, $NH_4HSO_4$, $NH_4NO_3$, and $NH_4IO_3$ (Hyttinen, 2023a) as well as in carboxylic acids (Hyttinen et al., 2020; Hyttinen and Prisle, 2020). The group contribution calculations in AIOMFAC can be finished in seconds, while the quantum chemistry calculations in COSMO-RS require hours to be completed. However, COSMO-RS is the only existing method able to estimate $\alpha_w$ in OS solutions, although requiring much more time for thermodynamic calculations than AIOMFAC. [...]

Section 3.1
[...] for all studied OS. Compared with the only existing measured $\alpha_w$ data from Bain et al. (2023), COSMO*therm* calculations provided similar values of $\alpha_w$ for NaMS and NaES (Fig. S2 in the Supplement).

[Figure]

. Figure 2. Hygroscopic growth factors (HGFs) for sodium methyl sulfate (NaMS; green), potassium glycolic acid sulfate (KGAS; yellow), and ammonium 2-hydroxyethyl-sulfonate (NH$_4$HES; purple) as a function of RH at 295 K. The HGF data from the literature (Estillore et al., 2016; Peng et al., 2021, 2022; Bain et al., 2023) and the COSMO*therm*-derived calculations are shown in solid circles and dashed lines, respectively.

[Figure]

. Figure S4. Hygroscopic growth factors (HGFs) of all OS particles as a function of RH at 295 K. The computed HGFs were estimated on the basis of the COSMO*therm*-derived water activities that used the FINE parameterization and are present in red dashed lines. The HGF measurement data from the literature are present in open circles in grey, with the best fit (black dashed lines) and 95% confidence intervals (95% CIs; shaded areas in grey).

[Figure]

. Figure S6. Hygroscopic growth factors (HGFs) of all OS particles as a function of RH at 295 K. The HGF derived from the COSMO*therm*-derived water activities are in dashed lines for FINE (red), TZVP (green), and ELYTE (blue) parameterizations. The HGF measurement data from the literature are present in filled circles in grey.

3. Lines 36-37: Is there any other successful usage of COSMO-RS, comparison between modeled and experimental water activities of other organic compounds indicating the appropriateness of COSMO-RS to be used for the prediction of HGF? A thorough explanation of this would enhance the scientific quality of the method.

Response: Previous thermodynamic studies using COSMO*therm* have provided $\alpha_w$ estimates consistent with those measured in bulk and particle phases for multifunctional atmospheric organics (Hyttinen et al., 2020; Hyttinen and Prisle, 2020). Unfortunately, there is no comparison between measured and COSMO*therm*-predicted HGFs. This is why one of the goals of our study is to predict the HGFs based on COSMO*therm*-predicted $\alpha_w$ and compare them with the existing measured HGFs. As the first study modeling HGFs based on the COSMO*therm*-estimated $\alpha_w$, our benchmarking study shows good agreement between COSMO*therm*-derived and experimental HGFs for most of the studied OS. This highlights the potential applicability of COSMO*therm* for estimating HGFs for other atmospheric compounds.

Change: Conclusion

[...] Previous thermodynamic studies using COSMO*therm* have provided $\alpha_w$ estimates consistent with those measured in bulk and particle phases for multifunctional atmospheric organics (Hyttinen et al., 2020; Hyttinen and Prisle, 2020). As the first study modeling HGFs based on the COSMO*therm*-estimated $\alpha_w$, we show good agreement between COSMO*therm*-derived and experimental HGFs in most cases of the studied OS. This highlights the potential applicability of COSMO*therm* for estimating HGFs for other atmospheric compounds in future works.

4. Lines 126-130: what can you conclude from this? Can you do some interpretation of this result? What's the relationship between this result and those in the following sections?

Response: Thank you for pointing out the need for more discussion about these results. We include the following sentences in section 3.1 and the conclusion.

Change: Section 3.1

[...] Moreover, for each studied OS, the order of cations is more or less the same, suggesting the presence of the Hofmeister effect. Sodium is usually enriched in sea spray particles (Salter et al., 2016), while potassium and ammonium are commonly found in continental aerosol particles influenced by biomass burning (Vasilakopoulou et al., 2023) and anthropogenic ammonia sources (Pai et al., 2021). Therefore, the $\alpha_w$ of OS in aerosol particles may vary depending on the influence of marine and continental air masses.

Conclusion

[...] In this study, the COSMO*therm* calculations show how the presence of sodium, potassium, and ammonium affects the $\alpha_w$ of OS. Given that these three cations originate from different emission sources in continental and marine environments, the $\alpha_w$ of OS can vary from one place to the next. [...]

5. Line 134: Is there any method to quantify the similarities between the model and the experimental results? If yes, do the same for other comparisons between model and experimental results. Doing this will enhance the scientific quality of the study and help the comparison with possible future studies.

Response: This is a very useful suggestion. To quantify the similarities between the observation data and model simulations, we now analyze their respective relative differences. We found that the relative differences between the measured and FINE-parameterized HGFs are mostly within $\pm 5\%$. Compared with the other two COSMO*therm* parametrizations, FINE overall shows the smallest relative differences from the observational data (Fig. R2).

In addition, we also add the following sentences in the main text for clarification and the comparison figure Fig. R2 as Fig. S5 in the Supplement.

[Figure]

. Figure R2. Relative differences between observed and model-simulated hygroscopicity growth factor for all OS particles at different RH at 295 K, with colors indicating FINE (red), TZVP (green), and ELYTE (blue) parametrizations.

**Change: Section 3.2**

[...] To evaluate the model performance for the studied OS particles, we compared the model simulations against the measurement data as indicated by the relative differences shown in Fig. S5 in the Supplement. There are different possible sources of uncertainties that may explain the discrepancy between experimental data and COSMO*therm* estimates. The uncertainty can originate from the errors in the experiments, the COSMO*therm*-estimated $\alpha_w$, or the assumptions made for calculating the HGF from $\alpha_w$. In this study, most of the discrepancies likely originate from the COSMO*therm*-estimated $\alpha_w$ of some of the ions. The original COSMO*therm* parametrization (Klamt et al., 1998) provided only poor estimates for water. The most recent COSMO*therm* parametrization performs much better and we found relatively good agreement between experimental data and our HGF calculations. Note that the COSMO*therm*-predicted HGFs overall agree very well with the measurement data, mostly showing relative differences of $\pm 5\%$ or less against the measured HGFs (Fig. S5). [...] Among the three COSMO*therm* parametrizations, FINE overall provided the best agreement across all the studied OS particles and showed the smallest relative differences against the measurement data (Fig. S5). Therefore, FINE was chosen for the detailed analysis here.

6. Line 136: "Note that the ... measurement data." The explanation here is quite vague.

Response: See response to comment No. 5.

7. Line 162: 1:3 $\longrightarrow$ 1:5?

Response: Thank you for pointing it out. Now we correct the typo.

[...] For the NaMS-AS mixture particles with 1:5 mass ratio, [...]

8. Sect. 4 Conclusions: I think it would be good to include a brief discussion of the performance of COSMOtherm with different cations.

135

Response: Now we include a brief discussion of the performance of COSMO*therm* with different cations. See response to comment No. 4.

**R2 Response to Reviewer 2**

Lie et al. unitized the COSMO0RS model to simulate the hygroscopicity growth factor of organic sulfate (OS) and organic sulfate and ammonium sulfate mixture. The study shows good agreement with some experimental results. This study has good potential to provide a way to better simulate hygroscopicity for climate models. However, I have some major questions that need further explanation from the authors.

Response: Thank you for your insightful comments. To address your comments, we provide the following responses.

**R2.1 General Comments**

1. Could you discuss why you see a discrepancy between the model-predicted and experiment-measured HGF in Figure S4? Also, I suggest moving Figure S4 to the main text since it is essential.

   Response: The COSMO*therm* calculations are purely based on quantum chemistry calculations. In other words, the calculations are not constrained by the observation data. The difference between the model-predicted and measured HGFs in Fig. S4 indicates the degree to which the COSMO*therm* model overestimates or underestimates the experimental data.

   We do not agree that moving Fig. S4 will enhance the readability. The key message from Fig. S4 is that among the three COSMO*therm* parametrizations, FINE overall provided the best predictions against the experimental data. As mentioned in the response to No.5 from reviewer 1, we have included the following sentences in the main text for clarification and the comparison figure Fig. R2 as Fig. S5 in the supplement.

   Change: Section 3.2
   [...] To evaluate the model performance for the studied OS particles, we compared the model simulations against the measurement data as indicated by the relative differences shown in Fig. S5 in the Supplement. There are different possible sources of uncertainties that may explain the discrepancy between experimental data and COSMO*therm* estimates. The uncertainty can originate from the errors in the experiments, the COSMO*therm*-estimated $\alpha_w$, or the assumptions made for calculating the HGF from $\alpha_w$. In this study, most of the discrepancies likely originate from the COSMO*therm*-estimated $\alpha_w$ of some of the ions. The original COSMO*therm* parametrization (Klamt et al., 1998) provided only poor estimates for water. The most recent COSMO*therm* parametrization performs much better and we found relatively good agreement between experimental data and our HGF calculations. Note that the COSMO*therm*-predicted HGFs overall agree very well with the measurement data, mostly showing relative differences of $\pm 5\%$ or less against the measured HGFs (Fig. S5). [...] Among the three COSMO*therm* parametrizations, FINE overall provided the best agreement across all the studied OS particles and showed the smallest relative differences against the measurement data (Fig. S5). Therefore, FINE was chosen for the detailed analysis here.

2. I am not fully convinced by section 3.3. First, NaMS measurements data shows a gradual increase of HGF with an increase of RH, but your model predicted HGF shows a step increase. I think the OS and ammonium sulfate (AS) mixture should be amorphous, which should not have a delinquency point. Moreover, do you have the hygroscopicity parameter of OS? I expect their hygroscopicity is lower than AS, and mixing AS and OS should have hygroscopicity lower than AS but higher than OS.

   Response: There is a misunderstnading. Here, the measurement data refer to the hygroscopic growth data of NaMS-AS and not to those of pure NaMS. A step-wise change is clearly observed in the measured HGF in the Fig. 7 of Peng et al. (2021) where the measurement data are from. Peng et al. (2021) suggest that the observed step-wise change in HGF indicates the full deliquescence of the studied particles. If the particles were in an amorphous state, we would not expect the step-wise change but instead gradual and continuous hygroscopic growth (Mikhailov et al., 2009).

   The step change in HGF is attributed to the full dissolution of AS, when it is mixed with certain organics (Choi and Chan, 2002; Chan et al., 2006; Peng et al., 2021, 2022). In other words, at RH below the observed step change in HGF,

AS may exist in a crystalline solid state or partially dissolve in the liquid phase. Similar to Hodas et al. (2016), which studied the HGF of polyethylene glycol oligomers mixed with AS, we assume that AS exists in solid-liquid equilibrium before reaching full deliquescence. To improve the clarity of the text, we improved the description in Section 2.4, as shown below.

Studying the hygroscopicity parameter ($\kappa$) is out of the scope of our paper. But Peng et al. (2021) have reported $\kappa$ values for NaMS and NaMS-AS under subsaturated conditions. Unlike your expectation, NaMS-AS has a lower $\kappa$ (0.454-0.495) compared to NaMS alone (0.537-0.604). Similar results are also observed in NaES, NaHMS and NaHES when they are mixed with AS (Peng et al., 2021, 2022).

Change: Section 2.4
[...] However, AS may not fully dissolve in aqueous mixtures under low RH conditions. Previous experimental studies show that when AS is mixed with certain carboxylic acids (Choi and Chan, 2002; Chan et al., 2006) and OS (Peng et al., 2021, 2022), a step change in HGF was typically observed at or below the DRH of AS, which is attributed to the full dissolution of AS. In other words, at RH below the observed step change in HGF, AS may exist in a crystalline solid state or partially dissolve in the liquid phase. For all the studied OS-AS mixture particles, step-wise changes in HGFs were observed in the measurements conducted by Peng et al. (2021, 2022). Similar to Hodas et al. (2016), which studied the HGF of polyethylene glycol oligomers mixed with AS, we assume that AS exists in solid-liquid equilibrium before reaching full deliquescence in the calculations. [...]

3. What size of particles you used in your model? Do you consider diffusion limit?

Response: COSMO*therm* is a thermodynamic model for estimating $\alpha_w$ in compounds of interest at vapor-liquid equilibrium. Diffusion is not considered in our calculations, the system is assumed to be in equilibrium at every RH. This is mentioned several times in the manuscript.

The model treats the OS particles as a bulk phase without considering the particular size. Since our HGF calculation neglects the Kelvin effect, the particle size has no effect on our calculation (see e.g., the end of Section 3.2). This can be also seen from equation (7) of the manuscript, where the masses of water and OS can be relative masses/mass fractions.

Change: Section 2.1
[...] was used to calculate activity coefficients of water under vapor-liquid equilibrium. [...]

Section 2.3
[...] This approach treats the OS particles as a bulk phase without considering particle size.

**R2.2 Specific Comments**

1. For equation 5, Have you considered the effect of volume loss due to molecular interactions (e.g., hydrogen bond) and packing efficiency (e.g., intermolecular space)?

Response: The uncertainty in particle volume expressed in Eqs. 4 and 5 depends on how accurate the density estimation is. In COSMO*therm*, the density of a mixture is estimated from the molecular volumes of individual species present in the mixture. This method does not consider intermolecular hydrogen bonding as a factor in the density calculation. However, a comparison between experimental and calculated densities (Hyttinen and Prisle, 2020) has shown that the approach used in COSMO*therm* produces accurate density estimates.

Change: Section 2.3
[...] In COSMO*therm*, $\rho_j$ is estimated from the molecular volumes of individual species present in the mixture, assuming close packing. Therefore, the formation of intermolecular H-bonds within the mixture is not considered in the density calculation.

2. Figure 1: do you have any explanation for why NaMS and NaES show a sigmoidal relationship?

Response: NaMS and NaES do indeed display a different shape for the $\alpha_w$ and $m_w$ relationship displayed in Figure 1. We extended the discussion about Figure 1 in Section 3.1 to provide more interpretation of the observed differences. Note however, that there is no underlying mathematical model for the $\alpha_w$ and $m_w$ relationship as COSMO*therm* is a thermodynamical model.

Changes: Section 3.1

[...] Here, COSMO*therm* predicts higher $\alpha_w$ in more concentrated solutions and lower $\alpha_w$ in more diluted solutions for NaMS and NaES, compared with those containing the isomeric hydroxy sulfonate salts (i.e., NaHMS and NaHES). This is likely caused by the relatively more hydrophobic nature of the methyl and ethyl groups compared to anions that contain hydrogen bond donating functional groups (e.g., -OH groups). Similar but stronger patterns in $\alpha_w$ are seen in the unstable regions of phase-separating mixtures (Hyttinen, 2023b).

3. Figure 2: Why do you only show NaMS, KGAS, and NH4HES?

Response: We intend to show the same level of details in Fig. 2 and Fig. S3 for easier comparison of the experimental and computed HGFs. Thus, we only depict NaMS, KGAS, and $NH_4HES$ in the main text figure to enhance the readability. These salts were selected to show one salt for each studied cation as examples in the main manuscript.

**References**

[revised manuscript text omitted]

---

## Author Response (AR2)

**R1    Response to Reviewer 1**

We thank Reviewer 1 for accepting the revised manuscript.

**R2    Response to Reviewer 2**

We appreciate Reviewer 2 for the comments. Below, we provide point-by-point responses to each comment. In the following text, the **reviewers' comments and suggestions** are in black, **authors' responses** are in red, and **changes to the manuscripts and supplement information** are in blue.

1. line 6: "We selected OS compounds for which the hygroscopic growth factors (HGF) have been experimentally studied." $\longrightarrow$ "The hygroscopic growth factors (HGF) of these OS compounds have been experimentally studied."

   Response: Thank you for pointing it out. We have modified the sentence accordingly.

2. line 19: "but also" $\longrightarrow$ "as well as"?

   Response: We have replaced "but also" with "as well as".

3. line 41-42: ", although requiring much more time for thermodynamic calculations than AIOMFAC" delete? because this has been mentioned in the previous sentence.

   Response: We have removed the sentence in the revised manuscript.

4. line 111: "assume that AS exists in solid-liquid equilibrium before reaching full deliquescence in the calculations." and line 124 "AS is assumed to exist only in its solid form." are they in conflict with each other?

   Response: Thank you for pointing it out. Now, we have corrected the sentence.

   Change:
   [...], we assume that AS exists only in the solid state before reaching full deliquescence in the calculations[...]

5. line 128? please add how you derived the HGF of mixtures when IAP<IAPsat.

   Response: We have included a brief description for the case of IAP<IAPsat.

   Change:
   [...] When the estimated $\mathrm{IAP} \leq \mathrm{IAP}_{sat}$, AS is assumed to be fully dissolved in the liquid phase and well mixed with OS. Similar to Eq. 7 which treats the particles as a bulk phase, the HGF in this case can be derived based on the COSMO*therm*-estimated $\alpha_w$ of the OS-AS mixture, with the aid of Eq. 10 expressed as below:

$$\mathrm{HGF} = \left( \frac{\frac{m_{\mathrm{OS}}}{\rho_{\mathrm{OS}}} + \frac{m_{\mathrm{AS}}}{\rho_{\mathrm{AS}}} + \frac{m_{\mathrm{H_2O},i}}{\rho_{\mathrm{H_2O}}}}{\frac{m_{\mathrm{OS}}}{\rho_{\mathrm{OS}}} + \frac{m_{\mathrm{AS}}}{\rho_{\mathrm{AS}}} + \frac{m_{\mathrm{H_2O},0}}{\rho_{\mathrm{H_2O}}}} \right)^{\frac{1}{3}} = \left( \frac{1 + \frac{m_{\mathrm{AS}} \cdot \rho_{\mathrm{OS}}}{m_{\mathrm{OS}} \cdot \rho_{\mathrm{AS}}} + \frac{m_{\mathrm{H_2O},i} \cdot \rho_{\mathrm{OS}}}{m_{\mathrm{OS}} \cdot \rho_{\mathrm{H_2O}}}}{1 + \frac{m_{\mathrm{AS}} \cdot \rho_{\mathrm{OS}}}{m_{\mathrm{OS}} \cdot \rho_{\mathrm{AS}}} + \frac{m_{\mathrm{H_2O},0} \cdot \rho_{\mathrm{OS}}}{m_{\mathrm{OS}} \cdot \rho_{\mathrm{H_2O}}}} \right)^{\frac{1}{3}} \tag{1}$$

   where $m_{\mathrm{AS}}$ and $\rho_{\mathrm{AS}}$ are the mass and density of AS.

6. line 201: "MS" $\longrightarrow$ "OS"?

   Response: Thank you for pointing it out. Now we correct the typo.